# Genetic and Metabolic Determinants of Atrial Fibrillation in a General Population Sample: The CHRIS Study

**DOI:** 10.3390/biom11111663

**Published:** 2021-11-09

**Authors:** David B. Emmert, Vladimir Vukovic, Nikola Dordevic, Christian X. Weichenberger, Chiara Losi, Yuri D’Elia, Claudia Volpato, Vinicius V. Hernandes, Martin Gögele, Luisa Foco, Giulia Pontali, Deborah Mascalzoni, Francisco S. Domingues, Rupert Paulmichl, Peter P. Pramstaller, Cristian Pattaro, Alessandra Rossini, Johannes Rainer, Christian Fuchsberger, Marzia De Bortoli

**Affiliations:** 1Eurac Research, Institute for Biomedicine (Affiliated to the University of Lübeck), 39100 Bolzano, Italy; David.Emmert@eurac.edu (D.B.E.); Vladimir.Vukovic@izjzv.org.rs (V.V.); Nikola.Dordevic@eurac.edu (N.D.); Christian.Weichenberger@eurac.edu (C.X.W.); Yuri.DElia@eurac.edu (Y.D.); Claudia.Volpato@eurac.edu (C.V.); Vinicius.Veri@eurac.edu (V.V.H.); Martin.Goegele@eurac.edu (M.G.); Luisa.Foco@eurac.edu (L.F.); Giulia.Pontali@eurac.edu (G.P.); Deborah.Mascalzoni@eurac.edu (D.M.); Francisco.Domingues@eurac.edu (F.S.D.); Peter.Pramstaller@eurac.edu (P.P.P.); Cristian.Pattaro@eurac.edu (C.P.); Alessandra.Rossini@eurac.edu (A.R.); Johannes.Rainer@eurac.edu (J.R.); 2Centre for Disease Control and Prevention, Institute of Public Health of Vojvodina, 21000 Novi Sad, Serbia; 3Department of Cardiology, Tappeiner F. Merano Hospital, 39012 Merano, Italy; Chiara.Losi@sabes.it (C.L.); Rupert.Paulmichl@sabes.it (R.P.); 4Department of Cellular, Computational and Integrative Biology—CIBIO, University of Trento, 38122 Trento, Italy; 5Centre for Research, Ethics and Bioethics Uppsala University, SE-751 05 Uppsala, Sweden

**Keywords:** atrial fibrillation, GWAS, rare alleles, familial aggregation, metabolomics, Cooperative Health Research in South Tyrol

## Abstract

Atrial fibrillation (AF) is a supraventricular arrhythmia deriving from uncoordinated electrical activation with considerable associated morbidity and mortality. To expand the limited understanding of AF biological mechanisms, we performed two screenings, investigating the genetic and metabolic determinants of AF in the Cooperative Health Research in South Tyrol study. We found 110 AF cases out of 10,509 general population individuals. A genome-wide association scan (GWAS) identified two novel loci (*p*-value < 5 × 10^−8^) around SNPs rs745582874, next to gene *PBX1*, and rs768476991, within gene *PCCA*, with genotype calling confirmed by Sanger sequencing. Risk alleles at both SNPs were enriched in a family detected through familial aggregation analysis of the phenotype, and both rare alleles co-segregated with AF. The metabolic screening of 175 metabolites, in a subset of individuals, revealed a 41% lower concentration of lysophosphatidylcholine lysoPC a C20:3 in AF cases compared to controls (*p*-adj = 0.005). The genetic findings, combined with previous evidence, indicate that the two identified GWAS loci may be considered novel genetic rare determinants for AF. Considering additionally the association of lysoPC a C20:3 with AF by metabolic screening, our results demonstrate the valuable contribution of the combined genomic and metabolomic approach in studying AF in large-scale population studies.

## 1. Introduction

In this work, we describe a systematic scan for genetic and metabolic determinants of AF. We conducted a GWAS of 19,057,004 markers in 10,518 individuals, familial aggregation analysis in the same set of individuals, and a screening of 175 metabolites in 5688 of the 10,518 individuals from the same general population study, where participants were unselected for any disease or condition.

Atrial fibrillation (AF) is a common cardiac arrhythmia that can cause severe stroke and heart failure and is associated with premature mortality [1]. AF is characterized by a progressive atrial remodeling, which results in electrical dissociation and local conduction heterogeneities, favoring re-entry and perpetuation of the arrhythmia, with irregular and often abnormally fast heart rate [2]. Multiple risk factors have been implicated for AF, including valvular heart disease, diabetes mellitus, coronary artery disease, chronic kidney disease, hypertension, obesity, and smoking [3]. However, AF is also known to occur in the absence of these conditions [4]. Family-based studies indicate that AF has a clear genetic predisposition [5,6]: the risk to develop AF is ~40% larger in first-degree relatives of an AF case [6]. Genome-wide association studies (GWAS) have identified >160 gene loci for AF [7]. Usually, common variants identified in GWAS occur in intergenic regions or in introns and are presumed to contain regulatory sequences that influence gene expression [7]. However, numerous rare coding variants have been identified in families and in early onset cases, predominantly in genes encoding cardiac ion channels, myocardial structural components, and cardiac transcription factors [7]. Cardiac excitation-contraction coupling and ion channel/pump integrity require oxygen and nutrients and are closely linked to cellular metabolic conditions. Recent studies describing the metabolite profiles of atrial tissues [8] and blood samples from AF patients [9,10,11] indicate a possible involvement of metabolic alterations in AF pathophysiology, but further efforts are necessary to fully explain heritability and metabolic processes in AF.

## 2. Materials and Methods

### 2.1. CHRIS Study and Phenotype Construction

The Cooperative Health Research in South Tyrol (CHRIS) study is a population-based study whose baseline assessment was carried out in Vinschgau/Val Venosta (Bozen/Bolzano, Italy) between 2011 and 2018 [12]. Participants were invited to the study centre located in the central town of the valley where they underwent biospecimen collection [13], interviews, and physical examinations including electrocardiographic (ECG) analysis using the PC-ECG-System Custo 200 (Customed) workstation with a sampling rate of 1000 Hz (Custo med GmbH, Ottobrunn, Germany). The 12-lead ECG at rest was registered as a conventional 10 s and a longer 20 min ECG recording, as described previously [14]. The values of P wave duration in milliseconds (ms) for each 10 s ECG were extracted and stored digitally. Collected blood samples were aliquoted, and serum and plasma samples were obtained for posterior biobanking. Additionally, information on medication use during the seven days prior to the study centre visit was collected by scanning the barcodes of drug boxes brought by the participants to the interview and classified using the Anatomical Therapeutic Chemical (ATC) coding system. Administration mode, frequency, and duration of therapy were registered for each scanned drug. Additional information pertinent to common risk factors for AF was collected. A description of data collection procedures is available in Appendix A.

We evaluated the collected data (CHRIS study data release 3.0) which include phenotype information from 10,518 study participants. The 10 s ECG recording was available for 10,438 (99.24%) individuals. Using clinical guidelines for the management of AF [2,3], we initially selected, as possible AF cases, participants who responded positively to the question “Do you have atrial fibrillation diagnosed by a doctor?” or who had a P wave duration of 0 ms at the 10 s ECG recording. To strengthen the selection, we further refined cases to those for whom we identified the use of cardiovascular drugs corresponding to the ATC classification group “C”, with specific dosage and information on administration mode, and consulted a clinical cardiologist for the experience-based identification of possible therapy for AF (Appendix A). In order to objectivise the method and minimize the misclassification of AF cases, 51 individuals who responded positively to the question “Do you have atrial fibrillation diagnosed by doctor?” but did not take the above-specified medications and had a P wave ≠ 0 ms were excluded as cases. All remaining participants not fulfilling the above phenotype classification criteria were defined as controls. Finally, nine participants were assigned as missing for the AF phenotype, since no information was available both for the 10 s ECG and for medication. This approach might lead to some false negatives in the large size control group; this, however, ensures having fewer false positives among AF cases.

### 2.2. Genome-Wide Association Study (GWAS)

The 10,518 individuals were genotyped in two batches using the Illumina Human OmniExpressExon and Omni2.5Exome chip arrays. Following quality control excluding samples with low call rate, poorly clustered variants, and samples found to be corrupt based on B allele frequency analysis, the two genotype datasets were merged and imputed based on the Haplotype Reference Consortium Panel (HRC) [15] dataset using the Michigan Imputation Server [16,17]. The final dataset included 10,147 samples and 19,057,004 SNPs.

GWAS of AF on the imputed allelic dosages with minimum allele count (MAC) of 3 was performed using the SAIGE_0.39 software package, which implements a generalized mixed model using saddlepoint approximation (SPA) and assuming additive genetic effects [18]. Statistical significance was assessed at the canonical genome-wide threshold of 5 × 10^−8^. To distinguish novel from known associations, a custom catalog of previously reported loci associated with AF was prepared for the SWISS analysis using NHGRI-EBI GWAS Catalog v1.0.2 [19], downloaded 20 February 2020. A subset of all GWAS Catalog records containing the term “fibrillation” was refined by filtering for GWAS Catalog “DISEASE_TRAIT” terms specific to atrial fibrillation and excluding records containing “DISEASE_TRAIT” terms indicating non-genetic etiology. Using this catalog, novelty of the identified variants was determined using the SWISS v1.1.1 package (https://github.com/statgen/swiss), with LD clumping set to r^2^ > 0.1, and distance-based clumping set at 1Mb. Statistical significance was assessed at the threshold of 5 × 10^−8^ and 5 × 10^−6^.

### 2.3. Familial Aggregation

We used the R/Bioconductor FamAgg package [20] to investigate if CHRIS participants with AF were aggregated within clusters of close relatives. We considered for this analysis all 10,518 CHRIS participants whose quality-controlled data were available on 26 May 2021; of these, 22 participants opted out from inclusion in genealogy analysis, one opted out from the study, and 815 participants did not report any relatives. This left 9680 participants for analysis, who were connected through a pedigree including 19,578 individuals, of whom 9672 had phenotype information available (8 were not phenotyped). Pairwise relatedness was expressed as pedigree-based kinship coefficients. According to the definition provided in paragraph 2.1, we observed in total 110 AF cases: 7 were singletons without reported ancestry, and 103 cases were used in the familial aggregation analysis. Analogously, 10,399 unaffected participants served as controls, and after removal of 807 singletons and 23 opt-outs from analysis in the control group, we arrived at 9569 controls and 103 cases, connected by 9898 un-phenotyped individuals. As discussed previously [21], in the context of large pedigrees, FamAgg’s Kinship Sum test represents the best method to detect familial aggregation as this test can quantitatively identify individuals with a significantly high number of affected kin pedigree members. Briefly, for a given case, this test takes the sum of kinship coefficients over all affected relatives and computes an empirical *p*-value derived from a background distribution resulting from 1,000,000 rounds of randomly assigning the 103 cases in the same set of pedigrees. Notice that the kinship sum statistics guarantees that the contributing cases are related by kinship and hence are sharing genetic material with each examined case. *p*-values were adjusted for multiple hypothesis testing utilizing the Benjamini–Hochberg method, and the false discovery rate was set at *p*-adj = 0.05. Pedigrees were examined and plotted using HaploPainter software v. 1.043 [22].

### 2.4. Sanger Sequencing

Selection of samples for Sanger sequencing was carried out by supplementing risk allele carriers identified with GWAS with potential carriers, defined as samples where the imputation genetic posterior probability (GPP) for carrier genotypes was >0.25. The GPP statistic, calculated by the minimac imputation algorithm used in the Michigan Imputation Server, represents the relative likelihood that a sample is actually each of the three possible genotypes of a given variant: homozygous reference: P(REF|REF); heterozygous: P(REF|ALT) or P(ALT|REF); or homozygous alternative: P(ALT|ALT). Controls to be sequenced were randomly selected from the remaining non-carriers.

Q5 High-Fidelity 2× Master Mix (New England Biolabs, Euroclone distributor, Milan, Italy) was used for DNA amplification following manufacturer’s instructions. The 4 PCR reactions were run on Mastercycler^®^pro S instrument (Eppendorf Italia, Milan, Italy) using the following conditions: 98 °C 30 s/98 °C 10 s; 67 °C 20 s; 72 °C 30 s for 30 cycles/72 °C 1 min; 4 °C 10 min. PCR products were purified using the QIAquick PCR Purification Kit (Qiagen Italia, Milan, Italy) and sent for sequencing at Eurofins Genomics (Ebersberg, Germany). Primers were designed using Primer3 tool v.4.1.0 (https://primer3.ut.ee/). Those used for amplification and sequencing are reported in Table 1.

### 2.5. Targeted Metabolomics

Metabolite concentrations in serum samples were quantified using the AbsoluteIDQ^®^ p180 kit from Biocrates (Biocrates Life Sciences AG, Innsbruck, Austria). In brief, 10 µL of serum samples from study participants, quality control samples (provided by Biocrates and pool of all serum samples), calibration curve samples, one double blank sample (Milli Q water), and one blank sample (PBS) were transferred to Biocrates filter plates using the Fluent 780 automatic pipetting system. Samples were further processed according to the manufacturer’s protocol, which included an initial filtering step followed by a derivatization employing phenyl isothiocyanate and sample extraction with 5 mM ammonium acetate solution in MeOH. Sample aliquots were then transferred into two distinct 96-well plates and filled with LC-MS and FIA-specific dilution solutions. Plates were then analyzed with a LC-MS system consisting of an Agilent UHPLC 1290 coupled with a Sciex 6500 QTRAP. For each sample, 42 metabolites (21 amino acids and 21 biogenic amines) were fully quantified by UHPLC-MS/MS using the calibration curve samples. Semi quantitative FIA-MS/MS was employed to assess the remaining metabolites concentration (40 acylcarnitines, 90 glycerophospholipids, 15 sphingolipids, and 1 sum of hexoses). For UHPLC-MS/MS data, integration of peaks was visually checked using Multiquant software v. 3.0.2 from SCIEX, and results were stored together with FIA-MS/MS data using the MetIDQ software v. 6.0.0 from Biocrates. Finally, metabolite raw concentrations were exported for subsequent data quality assessment and normalization, which was performed in R (version 4.0.3).

Between-plate differences were adjusted for each metabolite considering the concentrations measured for the QC samples per plate. For quality assessment, the coefficient of variation in QC samples, the number of missing values, and the signal distribution was considered. In total, 13 of the 188 metabolites were excluded because of poor quality or high number of missing values. For data imputation, missing values due to a signal below the detection limit of the instrument were replaced by random numbers from a uniform distribution ranging from the smallest measured value for that analyte to half of this smallest value.

Metabolomics concentrations were available for 5688 study participants, including 64 AF cases. To identify metabolites with different concentrations between AF cases and controls, linear regression models were fitted separately to each analyte using the log2 transformed concentration as a response variable and a binary variable for AF, sex, age, body mass index (BMI), self-reported fasting status, and a binary variable for AF medication as covariates.

In addition, linear regression models were fitted to metabolite concentrations standardized to mean of 0 and standard deviation of 1. Coefficients from this analysis, where differences in one unit are equal to a standard deviation of 1, are reported as “effect size”.

Raw *p*-values were adjusted for multiple hypotheses testing with Bonferroni correction.

Metabolites were considered significant if they had an adjusted *p*-value smaller than 0.05 and if, in addition, their difference in concentration was at least twice as large as the coefficient of variation (CV) of that metabolite. The CV was calculated on the QC samples representing the pool of all study samples, and it hence represents the technical variability observed for that metabolite in the present data set.

To evaluate a potential influence of medication on the concentrations of significant metabolites, we determined all ATC level 2 medications taken by more than 10 AF cases. These were “antithrombotic agents” (45 AF cases), “agents acting on the renin-angiotensin system” (33 AF cases), “beta blocking agents” (23 AF cases), “lipid modifying agents” (21 AF cases), “calcium channel blockers” (16 AF cases), “cardiac therapy” (14 AF cases), and “diuretics” (14 AF cases). After excluding AF cases, multiple linear regression models were fitted to the data with the log2 transformed metabolite concentration as response variable and participants’ age, sex, BMI, and self-reported fasting status as covariates.

### 2.6. Total Cholesterol, HDL and LDL

Total cholesterol, HDL, and LDL of all participants were quantified using clinically certified assays at the central laboratory of the Merano Hospital as described previously [13]. The same linear model than for the metabolomics analysis was applied to the log2 transformed data after excluding pregnant women and individuals treated with lipid modifying agents (77 AF cases, 9671 controls).

## 3. Results

### 3.1. AF Cases in CHRIS Study

A total of 97 (0.92%) participants had a P wave duration of 0 ms, and 103 (0.98%) responded positively to the question of whether they had been diagnosed with AF by a physician. For the current analyses, we selected 110 (1.05%) AF cases and 10,399 (98.87%) controls, while nine (0.09%) participants were assigned as missing. A total of 23 out of the 110 selected cases consumed specific medications in dose and mode most likely corresponding to AF therapy. Ten of the AF cases showed a persistent P wave = 0 ms while also taking specific medications. The 20 min ECGs available for 96 out of 97 AF cases, showing an absence of P wave, were manually checked by an expert. Sixty-nine (70.4%) individuals were confirmed to have a P wave = 0 ms, and seven (10.1%) of them showed atrial flutter.

Characteristics of AF cases and controls are shown in Table 2. Briefly, the mean age of the cases was 66.5 (SD = 19.7) years, and these were mostly males (60.9%), while controls had a mean age of 45.8 (SD = 16.2) years and were mostly females (55.4%). Fifty (45.5%) cases responded positively that they had been previously diagnosed with AF by a doctor. More than 88% of cases had a P wave = 0 ms at the 10 s ECG, and 76 (69.1%) participants used one or more of the drugs corresponding to the ATC classification group “C”.

### 3.2. Genome-Wide Association Study: Two Novel Loci Associated with AF

GWAS, for binary traits using SAIGE, detected four genome-wide significant associations (*p*-value < 5 × 10^−8^) of imputed variants for the AF trait in the HRC-imputed CHRIS genotype dataset (Table 3). Nine additional variants of borderline significance (*p*-value between 1 × 10^−6^ and 5 × 10^−8^) were also found and are reported in Appendix A.

Three of the genome-wide significant associations were located on chromosome 1 (Chr1), and one was located on chromosome 13 (Chr13) (Figure 1).

The four significantly associated variants were evaluated for novelty against previously reported GWAS signals using SWISS. Using *p*-value thresholds of 5 × 10^−8^ and 5 × 10^−6^, no matches to previously reported signals were discovered.

### 3.3. Familial Aggregation

Pedigrees in the CHRIS cohort exhibit a peculiar structure, where more than 90% of the participants are found in a single pedigree that horizontally connects extended families covering up to six generations by marry-ins. In particular, we have found 230 pedigrees. The largest pedigree, termed family XL, consisted of 8897 phenotyped participants connected by 9224 non-phenotyped individuals reported as relatives by their phenotyped cohort members. The remaining pedigrees are substantially smaller. The second largest pedigree contains 50 cohort members and, excluding family XL, the set of pedigrees is characterized by a median size of four members.

Familial aggregation analysis of the 103 cases and 9569 controls gave rise to 80 participants without further affected relatives in their extended families. Family XL contained 98 (95.1%) cases and therefore required usage of the Kinship Sum test to highlight affected individuals instead of computing a pedigree-wide score. We found seven extended families with two cases. We observed a parent–child relationship of cases once and found two pairs of affected siblings. The remaining four pairs of cases involved more remote kin. Finally, we discovered three extended families with three cases each, and one of them was reported with a significant *p*-value, *p*-adj = 2.94 × 10^−2^, which corresponds to affected participant III-2 with her affected son (IV-3) and brother (III-3) (Appendix A).

### 3.4. PBX1 and PCCA Genes in a Family with AF

For Sanger sequencing, 18 samples identified as putative carriers of at least one of the four imputed genome-wide significant variants were supplemented with an additional six samples selected as possible carriers based on their GPP. A total of 94 samples, chosen randomly from samples that had not been identified as carriers of any of the four variants of interest, were included as controls.

Overall, two out of the four genome-wide significant SNPs were confirmed by Sanger sequencing: rs745582874 next to the *PBX1* gene on Chr1, and rs768476991 within the *PCCA* gene on Chr13 (Figure 2).

Briefly, rs745582874 was confirmed in all 14 previously identified carriers and in an additional individual of the family identified by familial aggregation analysis (III-3, Figure 3 and Appendix A), who is an obligate carrier. Furthermore, rs768476991 was confirmed in three out of four previously identified carriers but also in an additional seven individuals: five of whom belong to the family just mentioned above (III-2, III-3, IV-2, IV-4, and IV-6, Figure 3 and Appendix A), and one of whom is a relative of an AF sporadic case carrying the same SNP, and one is a control.

Both of the confirmed SNPs are very rare. However, they show a threefold higher frequency in the CHRIS cohort (Table 3) than in the gnomAD database (European, non-Finnish), showing a MAF of 0.019% and 0.006%, respectively. Of note, the two risk alleles are enriched in the family identified by familial aggregation analysis, and both co-segregate with the phenotype in the three AF cases (Figure 3). The individual IV-3 (Figure 3) is a young man with a medical diagnosis of AF at the age of 31 years. During study enrollment, he showed aP wave = 0 at 10 s ECG. The other two AF cases, one man (III-3) with medical diagnosis of AF at the age of 68 years and a woman (III-2) of 78 years (without a reported AF diagnosis), showed a P wave = 0 at 10 s ECG (Figure 3), and both take specific cardio drugs. The woman also takes antithrombotic medication. We have also analyzed the 20 min ECGs for all family members, and only the three AF cases exhibited P wave = 0 along the entire 20 min. None of the other family members carrying the confirmed SNPs have AF signs.

Almost all AF cases carrying the confirmed variants belong to the family here described; however, an additional three sporadic AF cases were identified to be positive carriers. For a summary of demographic characteristics and common risk factors in carrier and non-carrier AF cases, see Appendix A.

### 3.5. Lysophosphatidylcholine lysoPC a C20:3, HDL, LDL, and Total Cholesterol Are Reduced in AF Cases

From all 175 tested metabolites, only lysophosphatidylcholine lysoPC a C20:3 showed a significant difference in concentrations between AF cases and controls (41% lower concentration in AF cases compared to controls; effect size: −1.075; adjusted *p*-value: 0.005) (Figure 4).

To investigate the potential relationship between lysoPC a C20:3 and medication, we first identified all ATC level 2 medication taken by at least 10 AF cases and subsequently tested for an influence of this medication on the metabolite’s concentration. No significant difference in concentration between treated and untreated controls was observed for any of the tested medications, suggesting that this metabolite is most likely independent of these treatments. We also found that in the family described above, lysoPC a C20: 3 is significantly lower in the nine available relatives (whether or not they have AF) (individuals III-2, III-3, III-4, IV-1, IV-3, IV-4, IV-5, IV-6, and IV-7, Figure 3) compared to the other CHRIS participants (24% lower concentration, *p*-adj = 0.031) (Figure 5). Interestingly, among family members, the two AF cases with a reported medically diagnosed AF (III-3 and IV-3, Figure 3) had the lowest values for lysoPC a C20: 3.

We also tested concentrations of HDL, LDL, and total cholesterol in blood samples of participants not being treated with lipid modifying agents. HDL showed a slight tendency of a lower concentration in AF cases (50.13 mg/dL vs. 53.97 in controls, *p*-value = 0.072), while both LDL and total cholesterol levels were significantly lower (LDL: 76.6 md/dL vs. 91.34 in controls, *p*-value = 0.0003; total cholesterol: 146.39 mg/dL vs. 162.47, *p*-value = 0.0014).

## 4. Discussion

The CHRIS study constitutes a valuable resource to investigate the genomic, metabolomic, and environmental factors in human diseases. The low residential mobility across generations and low inbreeding, with a rather homogeneous lifestyle and environmental conditions, are elements that facilitate the identification of specific enriched genetic variants and altered metabolites in the CHRIS cohort [12]. On the other hand, it is important to consider that the study involves a small region; therefore, any possible findings cannot be generalized and need confirmation in other populations. Moreover, the fact that the CHRIS study is not a cohort of patients, with only limited clinical data available and with only a subset of cases showing the phenotype of interest, could be a limiting factor for the type of study presented here. A general population is more suitable for the study of common variants associated with frequent phenotypes affecting many individuals. However, a general population such as that we have in the CHRIS study, consisting of families with several members and even generations residing in an area characterized by low regional migration, may help to also identify rare variants associated with a specific phenotype. The current work is an example of how AF can be investigated in a general population study context, considering the contribution of genomics and metabolomics.

AF is the most common cardiac arrhythmia with a reported prevalence of approximately 3% in the global population [2], ranging from 0.56% previously reported in Japan [23] up to 6.1% and 5.5% in some studies from Spain [24] and the Netherlands [25], respectively. Other European studies described similar results [26,27,28]. The differences in reported prevalence may reflect distinct study settings, methods, and criteria used to identify AF cases and also differing characteristics of the study populations such as genetics, age, and ethnicity, as well as the presence of other heart diseases, risk, and environmental factors [29,30]. Here, we report 110 (1.05%) AF cases identified among the study participants, using their answers from the questionnaire and the results from 10 s ECG recordings, which are likely to identify those already having a confirmed AF diagnosis or, in case of a lack of this information, only those with a permanent AF, identified using a single-occasion ECG. We may conjecture that our current result is liable to underestimate the real prevalence of AF in our study sample, failing to detect those subclinical, silent, and paroxysmal AF cases that would remain unrecognized using this approach. Additionally, we also have to consider the specific AF drug therapy, which itself is limited due to questions of compliance, as a further limiting factor for a more reliable selection of AF phenotype.

In order to understand the genetic contribution to our AF cases, we performed a GWAS analysis and subsequently sequenced the identified four genome-wide significant variants. Given the low frequency of identified variants, Sanger sequencing validation was necessary since all variants were imputed. While the HRC panel used in imputation was expected to enable imputation of low-frequency and rare variants in European populations [31], such as the CHRIS cohort, extremely low minor allele frequency, as observed for the unconfirmed variants, is known to impede imputation accuracy [32]. This was reflected in the low imputation R^2^ values (<0.5) for the two unconfirmed variants, which indicates that very low confidence was achieved in their imputation. The fact that subsequent Sanger sequencing identified additional carriers, initially imputed as non-carriers, and that two out of four SNPs were not confirmed, demonstrates that this approach may recover missed carrier samples of poorly imputed variants and is fundamental to confirm the presence of SNPs in the studied cohort when the imputation quality is not very reliable.

The two Sanger-confirmed SNPs have never been reported to be associated with AF, not even in recent studies using very large AF sample sizes with high statistical power [33,34,35]. Their identification in the CHRIS cohort may have been facilitated, as these two SNPs are both enriched in a family where they segregate with the AF phenotype. Specifically, in this family, three out of five individuals carrying both SNPs are AF cases. Two of them are men with a reported medical AF diagnosis, and one is an elderly woman showing an absence of P wave and taking specific cardiac drugs. On the other hand, the two carriers without AF signs are two young women. However, we cannot exclude that these two young individuals may eventually develop AF later.

Interestingly, the two significant SNPs are respectively close or within the probable candidate genes, *PBX1* and *PCCA,* both of which are expressed in atria. We disregarded the *LMX1A* gene since its function is mostly related to the development of dopamine-producing neurons during embryogenesis. *PBX1* acts as a transcription factor and has an important role during cardiac development, specifically regulating distinct transcriptional pathways to control great-artery patterning and cardiac outflow tract septation [36]. Mutations in this gene have recently been associated with CAKUTHED (congenital anomalies of kidney and urinary tract with or without hearing loss, abnormal ears, or developmental delay) syndrome, characterized by multiple congenital defects including congenital heart disease [37]. The involvement of other genes with a specific function in the cardiac development and associated with AF was previously reported. For example, in a recent study, which has identified 151 genes associated with AF [35], pathway and functional enrichment analyses have highlighted processes related to cardiac development as being functionally relevant in AF pathogenesis. Of note, in a genome-wide whole blood transcriptomic analysis from the Framingham Heart Study, the expression of *PBX1* gene was found to be up regulated in AF patients [38].

The *PCCA* gene encodes the mitochondrial propionyl-CoA carboxylase, and mutations in this gene lead to an enzyme deficiency resulting in propionic acidemia, often associated with long QT syndrome [39]. It is interesting to highlight that long-QT syndrome has also been associated with AF [40,41], apparently by inducing early afterdepolarization-mediated ectopic activity [42]. Moreover, cardiac dysfunction in Pcca-/-(A138T) mice was associated with lower systolic Ca^2+^ release, impairment in the sarcoplasmic reticulum Ca^2+^ load, and decreased Ca^2+^ re-uptake by SR-Ca^2+^ ATPase (SERCA2a) [43]. Notably, it has been reported that Ca^2+^-handling dysregulation can promote beat-to-beat alternation in action potential duration, favoring reentry and causing AF [44]. Indeed, Ca^2+^ handling abnormalities are a common finding in atrial cardiomyocytes of AF patients [45,46,47]. These findings indicate that *PBX1* and *PCCA* could be considered very good candidate genes for AF.

In the last decade, numerous studies have been performed to better understand the genetic determinants of AF, but there remains scant literature on the role of altered metabolic pathways in AF. To identify metabolites discriminating between AF cases and controls, we performed targeted metabolomic analysis which allowed us to find a significantly lower concentration of the lysophosphatidylcholine lysoPC a C20:3 in AF cases. Tests performed for commonly prescribed ATC level 2 medications of AF cases suggest this difference being most likely independent of these treatments, even if drug intake and dosage were not considered. This result is consistent with previous data reporting altered lipid metabolites and, in particular, a minor amount of lysoPC a C20:3 in AF patients [11]. Moreover, in a recent analysis [48] conducted in a related population-based study from the same region, we observed an association between higher levels of another phosphatydilcholine, the PC38:3, and longer P wave duration, which is also a risk factor for AF [49]. Even though the association was replicated in an independent population sample from Scotland, proving its robustness, mediation and Mendelian randomization analyses showed that the association between PC38:3 and P wave was caused by the confounding role of BMI: in fact, BMI was proven to cause both PC38:3 increase and P wave prolongation [48]. Thus, any association between lipids and AF or its markers should be considered carefully, until proof of causality is given.

Since in blood, significant amounts of lysophosphatidylcholines are formed together with cholesterol esters by a specific enzyme system, lecithin cholesterol acyltransferase (LCAT), we decided to also test the LDL, HDL, and total cholesterol levels. LDL and total cholesterol were significantly lower in AF cases compared to controls, and HDL also showed a tendency of a lower concentration in AF cases. Of note, in a recent paper, a significant reduced activity of the LCAT enzyme in AF patients was also described [50], and it has been reported that LCAT could modify oxidative and inflammatory processes, as supported by an inverse relationship with the anti-oxidative, anti-inflammatory, and anti-thrombotic functions of HDL [50,51]. Several works have indicated that cardiomyocyte NACHT, LRR, and PYD domains containing protein-3 (NLRP3)-inflammasome signaling play a causative role in AF development [52,53,54,55]. Notably, the expression of NLRP3 and its downstream inflammatory cytokines is reduced in the presence of HDL [56], and it appears that the inhibition of NLRP3 inflammasome is due to the blocking of the serum amyloid A activity caused by HDL [57]. Taking into account all the above evidence, we can speculate that the lower concentration levels of lysoPCa C20:3 and HDL found in AF cases could have a role in the progression of AF through the NLRP3 inflammasome activation. The lower amount of lysophosphatidylcholine and cholesterol could be caused by a decreased LCAT activity with a subsequent NLRP3 activation. This hypothesis, of course, deserves further investigations, which are, however, beyond the scope of this work and should be more properly carried out in large cohorts of AF patients.

## 5. Conclusions

While results warrant replication in an independent sample, previous evidence on *PBX1* and *PCCA* genes and family segregation of the risk alleles with AF indicates that the loci identified by the rs745582874 and rs768476991 SNPs could be considered novel genetic rare determinants for AF. The additional identification of lower blood levels of lysoPC a C20:3 in association with AF underlines the value not only of the genomic approach but also of the metabolomics screening to study AF in large-scale population studies.

## Figures and Tables

**Figure 1 biomolecules-11-01663-f001:**
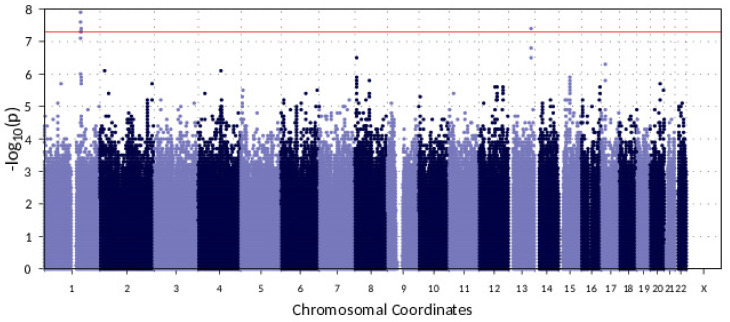
Manhattan plot. Red line indicates genome-wide significance level of 5 × 10^−8^. X: Chromosome X (not included in the GWAS). See Appendix A for the QQ plot.

**Figure 2 biomolecules-11-01663-f002:**
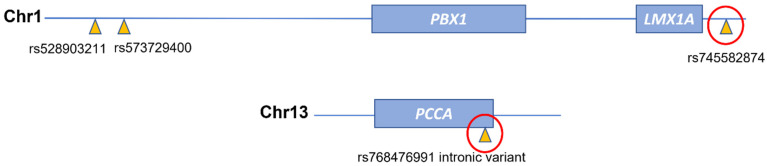
Schematic representation of genomic localization of 4 significant SNPs. The red circles indicate the confirmed SNPs by Sanger sequencing.

**Figure 3 biomolecules-11-01663-f003:**
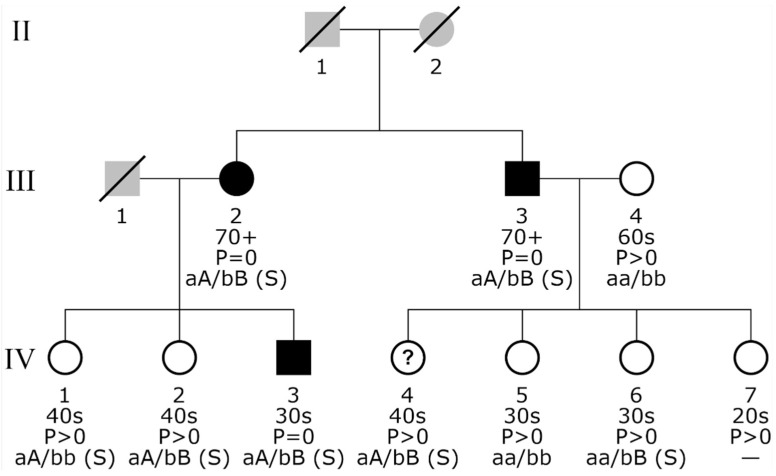
Minimal pedigree containing investigated AF cases. For clarity, the pedigree starts with the second generation and displays only the successors of persons with AF; a full pedigree spanning up to 5 generations is provided in Appendix A. Gray symbols indicate known, non-participating, and therefore not phenotyped individuals without further information. The three AF cases are highlighted by black symbols, whereas white symbols refer to unaffected study participants. Age groups (in decades) are provided in the second line below each individual: 20s, age 18 to 29 years; 30s, age 30 to 39 years; etc.; 70+, 70 years or older. The third line presents information on the 10 s ECG P wave, which is either zero (P = 0) or not zero (P > 0). The last line provides detailed genotype information for the loci of interest. Homozygous carriers of the reference allele of variant rs745582874 on chromosome 1 are shown as “aa”, and heterozygous carriers of this variant are marked by “aA”. The same type of information is encoded for rs768476991 on chromosome 13 using the letters “bb” and “bB”, respectively. Genotype information refers to the HRC-imputed data set unless followed by an additional “(S)”, which denotes confirmation by Sanger sequencing. A hyphen (“-“) is displayed for missing genotype information. Persons with a question mark (“?”) inside their symbol have an incomplete phenotype characterized by missing information on drug intake.

**Figure 4 biomolecules-11-01663-f004:**
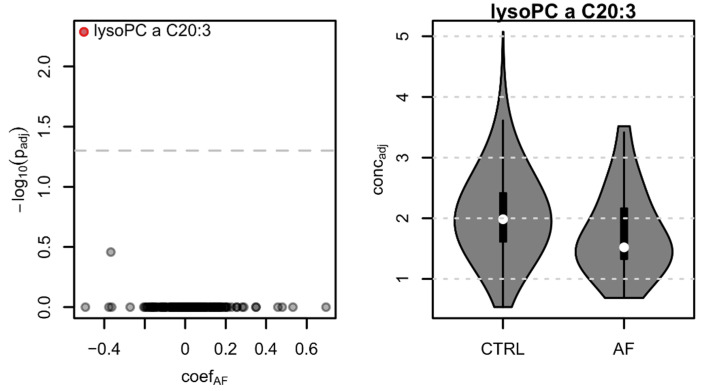
Left: Volcano plot representing the results from the targeted metabolomics analysis. coef_AF_: log2 difference of metabolite concentrations between AF cases and controls. Significant metabolites are highlighted in red. Right: lysophosphatidylcholine a C20:3 concentration (adjusted for age, sex, BMI, and AF therapy) in controls (CTRL) and AF cases.

**Figure 5 biomolecules-11-01663-f005:**
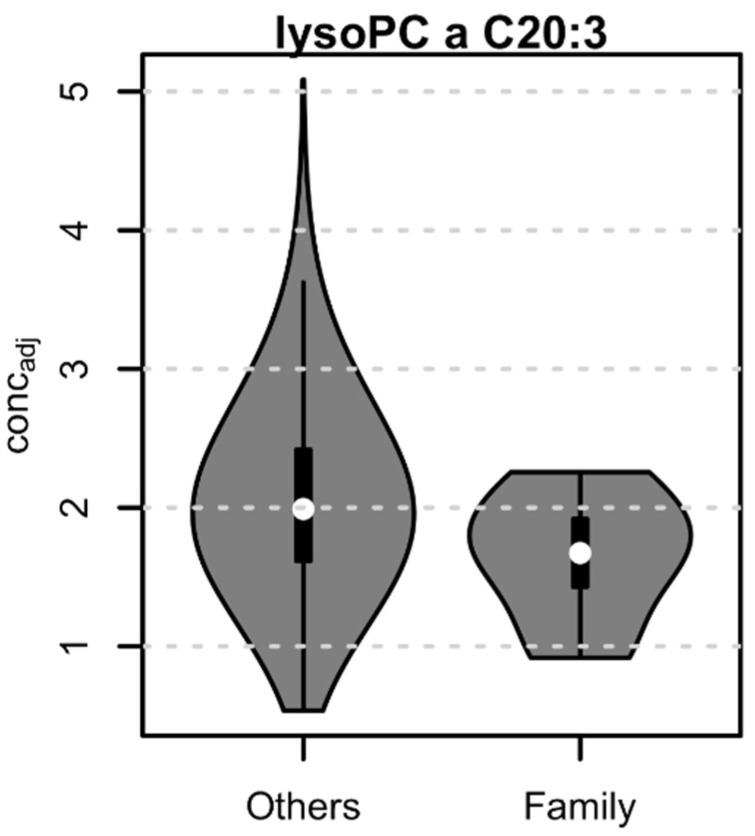
Lysophosphatidylcholine a C20:3 concentration (adjusted for age, sex, BMI and AF therapy) in nine family members and other CHRIS participants.

**Table 1 biomolecules-11-01663-t001:** PCR and sequencing primers.

Primer Name	Sequence 5′–3′	Used for Sequencing
rs528903211-F	GCCCAGACCGTGAAAACATT	
rs528903211-R	GGCATTTCGTCTCTGCCCTAG	Yes
rs573729400-F	ACAATCTGATTTGCCCAGCG	
rs573729400-R	CATGGGATGGGTCAAATTCTATGC	Yes
rs745582874-F	CCACTGATGATGGCCTTGG	Yes
rs745582874-R	GACATGGTCCTAGAGCTGCT	
rs768476991-F	TCTGTTCACACGGCTTAGATGA	Yes
rs768476991-R	GGTAATCCAACACTGTCCAAACA	

F: forward; R: reverse.

**Table 2 biomolecules-11-01663-t002:** Demographic characteristics and common risk factors in AF cases and controls.

	Cases (n = 110)			Controls (n = 10,399)			
	Missings, n (%)	Mean	Min/Max	Missings, n (%)	Mean	Min/Max	*p*-Value
		(SD)			(SD)		
**Age at participation—years**	0	66.51	18/93	0	45.78	18/94	<0.001
		(19.66)			(16.1)		
**Sex, n (%)**							
male	67 (60.9%)			4642 (45%)			0.001
female	43 (39.1%)			5757 (55%)			
**Do you have atrial fibrillation diagnosed by doctor? n (%)**							
yes	50 (45.5%)			51 (0.5%)			<0.001
no	60 (54.5%)			10,334 (99.4%)			
missing	0			14 (0.1%)			
**P wave (ms)**	0	13.42	0/150	71 (0.7%)	105.93	46/208	<0.001
		(37.46)			(12.49)		
**P wave = 0, n (%)**							
yes	97 (88.2%)			0			<0.001
no	13 (11.8%)			10,328 (99.3%)			
missing	0			71 (0.7%)			
**Medication corresponding to the ATC classification group “C”, n (%)**							
yes	76 (69.1%)			1590 (15.3%)			<0.001
no	34 (30.9%)			8420 (81%)			
missing	0			389 (3.7%)			
**Body Mass Index—kg/m2**	5 (4.5%)	27.83	18.3/40.2	65 (0.6%)	25.82	15.5/55.1	<0.001
		(4.69)			(4.6)		
**Systolic BP,** **mean of 3 measurements—mm/Hg**	5 (4.5%)	131.45	92/175	273 (2.6%)	122.16	81/226	<0.001
		(18.5)			(16.37)		
**Diastolic BP,** **mean of 3 measurements—mm/Hg**	5 (4.5%)	81.13	59/122	273 (2.6%)	78.26	51/145	0.002
		(11.43)			(9.32)		
**Cholesterol total—mg/dL**	0	196.98	95/287	6 (0.06%)	211.26	87/586	0.0004
		(38.79)			(41.72)		
**Cholesterol** **High Density Lipoprotein (HDL)—mg/dL**	0	56.62	27/92	7 (0.07%)	61.13	20/147	0.0028
		(15.35)			(15.78)		
**Cholesterol** **Low Density Lipoprotein (LDL)—mg/dL**	0	120.14	31/220	8 (0.08%)	131.42	24/332	0.0016
		(35.71)			(37.27)		
**C-Reactive Protein (CRP)—mg/dL**	0	0.42	0.02/2.85	5 (0.05%)	0.27	0/17.24	0.002
		(0.49)			(0.48)		
**Creatinine—mg/dL**	0	1.02	0.68/2.29	6 (0.06%)	0.88	0.36/8.69	<0.001
		(0.26)			(0.17)		
**eGFR—mL/min/1.73 m^2^**	0	73.24	26.06/134.25	7 (0.07%)	91.9	6.19/137.17	<0.001
		(21.97)			(16.12)		
**Smoking habit, n (%)**							
never	60 (54.5%)			5540 (53.3%)			0.042
past	39 (35.5%)			2975 (28.6%)			
current	10 (9.1%)			1836 (17.6%)			
missing	1 (0.9%)			48 (0.5%)			
**Has a doctor ever said that you have high blood pressure or hypertension? n (%)**							
yes	56 (51%)			2337 (22.5%)			<0.001
no	53 (48%)			8006 (77.0%)			
missing	1 (0.9%)			56 (0.5%)			
**Do you have diabetes mellitus? n (%)**							
yes	11 (10.0%)			269 (2.6%)			<0.001
no	98 (89.1%)			10,111 (97.2%)			
missing	1 (0.9%)			19 (0.2%)			
**Has a doctor ever told you that you have a heart failure? n (%)**							
yes	17 (15.4%)			50 (0.4%)			<0.001
no	86 (78.2%)			10,286 (99.0%)			
missing	7 (6.4%)			63 (0.6%)			
**Have you ever been told by a doctor that you had a stroke? n (%)**							
yes	13 (11.8%)			88 (0.7%)			<0.001
no	96 (87.3%)			10,278 (99.0%)			
missing	1 (0.9%)			33 (0.3%)			
**Have you ever been told by a doctor that you had a myocardial infarction? n (%)**							
yes	9 (8.2%)			118 (0.1%)			<0.001
no	98 (89.1%)			10,257 (98.6%)			
missing	3 (2.7%)			24 (0.3%)			

n = number; SD = standard deviation; BP = blood pressure; eGFR = estimated glomerular filtration rate. T-test was used for continuous variables and Fisher’s exact test for categorical variables.

**Table 3 biomolecules-11-01663-t003:** AF genome-wide significant variants.

Chr	POS (GRCh37)	REF/ALT	rsID	MAF	MAC	*p*-value	R2
1	164062528	C/G	rs528903211	0.00050	10	1.1 × 10^−8^	0.47
1	164110550	G/A	rs573729400	0.00056	9	2.4 × 10^−8^	0.43
1	165331300	T/G	rs745582874	0.00066	13	4.3 × 10^−8^	0.81
13	101142909	C/T	rs768476991	0.00028	5	4.1 × 10^−8^	0.48

Chr: chromosome; POS: genomic position; REF/ALT: reference and alternative alleles; MAF: minor allele frequency in CHRIS; MAC: minor allele count in CHRIS; R2: imputation quality index; value is between 0 and 1, with 1 indicating the highest level of certainty of imputation.

## Data Availability

The data presented in this study are available on request from the corresponding authors.

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
