# Peer review of "Genetic and Metabolic Determinants of Atrial Fibrillation in a General Population Sample: The CHRIS Study"

_biomolecules, 2021, doi:10.3390/biom11111663_

Round 1
Reviewer 1 Report
This is an interesting study involving a large CHRIS cohort in Italy with low population migration. The authors studied >1000 individuals in a GWAS for AF and followed by genealogy, extended pedigree, DNA sequencing and metabolomics analysis.
They reported 2 new loci reaching genome wide significance at or near to PBX1 and PCCA genes. Both at risk alleles are rare alleles with MAF of less than 1 in a thousand.
In extended pedigree study, potential segregation was found.
This study demonstrated the difficulties of study of rare alleles for phenotypes with incomplete penetrance. By using the approaches undertook by the authors, they finally demonstrated the 2 loci were statistically significant.
There is one major concern to be address:
Why other known AF loci did not show up in the study ?
It would be more re-assuring if they were also found (though may be with borderline significance).
The authors may list those loci with SNP reaching 1e-6 as other potential GWAS hits. It could also be beneficial for future follow up studies.
Reviewer 2 Report
The authors investigated the association of genetic and metabolic components with atrial fibrillation (AF) from the CHRIS study. They identified two loci mutations near to gene PBX1 and PCCA and the reduced concentration of lysophosphatidylcholine lysoPC in AF cases. It is an excellent effort for the results and congratulation to the authors. There were several points from the view of the clinical physician.
- Usually, the incidence of AF increases with aging, and age is the most significant risk factor for the development of AF for clinicians. To ensure the genetic contribution to AF, I recommend the comparison of the gene (+) and (-) AF patients in Table.
- In the definition of AF, the author described that they extracted the P wave duration digitally. It seemed that the author did not confirm from the expert. Please clarify the definition, and if the ECG did not confirm by the expert, it is the limitation. And are there any patients with atrial flutter?
- The genetic study is from a small region, so that the generalization could be limited.
Minor comments
- The last sentence of the introduction seems to be a conclusion, and I think it needs to remove.
- The sentence at the 242~244 lines is hard to understand and causes some confusion.
- In Table 2, the author should compare two groups and show the P-value of the test.
Reviewer 3 Report
Atrial fibrillation (AF) is a common cardiac arrhythmia that can cause severe stroke and heart failure and is associated with premature mortality. This manuscript is dedicated to expanding the limited understanding of AF biological mechanisms. By investigating the genetic and metabolic determinants of AF in the CHRIS study, the authors identified two loci (rs745582874 and rs768476991 SNPs) that could be considered novel genetic rare determinants for AF, along with additional identification of lower blood levels of lysoPC a C20:3 in association with AF. These findings underline the value not only of the genomic approach but also of the metabolomics screening to study AF in large-scale population studies. My suggestion - publish in present form.
Author Response
We thank the Reviewer for the positive comment about the paper.